Bayesian estimation of rainfall dispersion in Thailand using gamma distribution with excess zeros

Khooriphan Wansiri
Niwitpong Sa-Aat sa-aat.n@sci.kmutnb.ac.th
Niwitpong Suparat
Department of Applied Statistics, Faculty of Applied Science, King Mongkut’s University of Technology North Bangk , Bangkok , Thailand
Kuriqi Alban
Electronic publication date: 2022 Sep 16
Publication date: 2022
Volume: 10
Electronic Location ID: e14023
Received 2022 Jun 1; Accepted 2022 Aug 16
Copyright: ©2022 Khooriphan et al.
Copyright year: 2022
Copyright holder: Khooriphan et al.
License: This is an open access article distributed under the terms of the Creative Commons Attribution License, which permits unrestricted use, distribution, reproduction and adaptation in any medium and for any purpose provided that it is properly attributed. For attribution, the original author(s), title, publication source (PeerJ) and either DOI or URL of the article must be cited.
License URL: https://creativecommons.org/licenses/by/4.0/

Keywords: Bayesian estimation, Variance of a gamma distribution with excess zeros, Jeffrey’s prior, Uniform prior, Normal-gamma-beta prior, Rainfall dispersion, Fiducial quantity

Funding: National Science, Research, and Innovation Fund (NSRF) King Mongkut’s University of Technology North Bangkok KMUTNB–FF–66–44 This research received financial support from the National Science, Research, and Innovation Fund (NSRF), and King Mongkut’s University of Technology North Bangkok (Grant No. KMUTNB–FF–66–44). The funders had no role in study design, data collection and analysis, decision to publish, or preparation of the manuscript.

==============================
The gamma distribution is commonly used to model environmental data. However, rainfall data often contain zero observations, which violates the assumption that all observations must be positive in a gamma distribution, and so a gamma model with excess zeros treated as a binary random variable is required. Rainfall dispersion is important and interesting, the confidence intervals for the variance of a gamma distribution with excess zeros help to examine rainfall intensity, which may be high or low risk. Herein, we propose confidence intervals for the variance of a gamma distribution with excess zeros by using fiducial quantities and parametric bootstrapping, as well as Bayesian credible intervals and highest posterior density intervals based on the Jeffreys’, uniform, or normal-gamma-beta prior. The performances of the proposed confidence interval were evaluated by establishing their coverage probabilities and average lengths via Monte Carlo simulations. The fiducial quantity confidence interval performed the best for a small probability of the sample containing zero observations (δ) whereas the Bayesian credible interval based on the normal-gamma-beta prior performed the best for large δ. Rainfall data from the Kiew Lom Dam in Lampang province, Thailand, are used to illustrate the efficacies of the proposed methods in practice.

Introduction

Thailand is a mainly agrarian country, with the largest agricultural area being in the north of the country due to its cooler climate making it the best place for cultivation. Rainfall is an important factor for cultivation. The rainy season begins in mid-May and ends in mid-October, the southwest monsoon predominate over Thailand to bring abundant annual rainfall. August to September is the wettest period of the year for most of the country, whereas January and December are very dry. Fluctuating rainfall makes it difficult to predict heavy precipitation that could lead to crop loss or damage. Since environmental data, meteorology, climatology and pollution studies are often right-skewed, the gamma distribution is commonly used to model these data (Piao & Zhi-Sheng, 2015; Pradhan & Kundu, 2011; Son & Oh, 2006; Wang et al., 2019). Many researchers have developed confidence intervals for the parameters of a gamma distribution by using various methods. For example, Krishnamoorthy & León-Novelo (2014) proposed the parametric bootstrap (PB) confidence interval for the mean of a gamma distribution that performed satisfactorily even for small samples. Sangnawakij, Niwitpong & Niwitpong (2015) proposed the method of variance estimates recovery (MOVER) and score and Wald intervals to construct confidence intervals for the ratio of the coefficients of variation (CVs) of gamma distributions that performed better than classical estimators in terms of the expected length. Krishnamoorthy & Wang (2016) developed approximate fiducial quantities (FQs) for constructing the confidence interval for the mean of a gamma distribution that performed satisfactorily when the shape parameter was around 0.5 or larger. FQs can be used to establish approximate solutions to many statistical problems and can be readily applied to handle both uncensored and censored samples. Wang et al. (2019) proposed FQs for the differences between the shape parameters, scale parameters, and means of two independent gamma distributions and found that the performances of the FQ-based confidence intervals were more accurate than other comparable methods.

Rainfall data often contain zero observations at certain times of the year and so this must be taken into account when studying precipitation in Thailand. Aitchison (1955) investigated situations where data contain zero observations with the probability of 0¡ δ¡1 while the positive observations have a residual probability of 1- δ. Aitchison & Brown (1963) introduced the delta-lognormal distribution (a lognormal distribution with an excess of zero observations) for which the number of zero observations comprises a random variable with a binomial distribution and the positive observations comprise a random variable from a lognormal distribution. Many researchers have developed methods to construct confidence intervals for the parameters of a delta-lognormal distribution by using various methods. For example, Yosboonruang, Niwitpong & Niwitpong (2019) proposed new confidence intervals for the CV of a delta-lognormal distribution by using Bayesian methods based on the independent Jeffreys’, Jeffreys’ rule, or uniform prior and compared them with the fiducial generalized confidence interval (FGCI); the Bayesian confidence interval based on the independent Jeffreys’ prior performed better than the other methods in all situations studied. Maneerat & Niwitpong (2021) proposed confidence intervals for the common mean of several delta-lognormal distributions based on FGCI, the large-sample (LS) approach, MOVER, PB, and highest posterior density intervals (HPD) based on the Jeffreys’ rule (HPD-JR) or normal-gamma-beta (HPD-NGB) prior; those based on MOVER and PB outperformed the others in a variety of situations. Several researchers have examined methods for constructing confidence intervals for a gamma distribution with excess zeros. Ren, Liu & Pu (2021) proposed simultaneous confidence intervals for the difference between the means of multiple zero-inflated gamma distributions by using three fiducial methods and applied them to precipitation data. Muralidharan & Kale (2002) defined a modified gamma distribution with a singularity at zero and produced confidence intervals for the mean of a mixed distribution. Lecomte et al. (2013) provided compound Poisson-gamma and delta-gamma distributions to handle zero-inflated continuous data under variable sampling volume. Kaewprasert, Niwitpong & Niwitpong (2022) proposed Bayesian estimation for the mean of delta-gamma distributions with application to rainfall data in Thailand.

In statistics, the variance, which gives a measure of the spread or variability of a distribution, is the second central moment, and the positive square root of the variance is the standard deviation (Casella & Berger, 2001). It is one of the most popular parameters of interest for probability and statistical inference.

We are interested to study the confidence interval for the variance of gamma distribution because it is commonly used to model environmental data such as a rainfall dispersion. Rainfall dispersion data can help to examine rainfall intensity, which may be high or low risk. We have studied many research related to constructing the confidence interval for rainfall data, such as Yosboonruang, Niwitpong & Niwitpong (2019) and Maneerat & Niwitpong (2021). We have found several interesting priors, including: Jeffreys’, uniform, or normal-gamma-beta prior. Therefore, we applied to this study.

Since no publications have yet been forthcoming on constructing confidence intervals for the variance of a gamma distribution with excess zeros, the objective of the present study is to construct the confidence interval for the variance of a gamma distribution with excess zeros based on FQ, PB, and six Bayesian-based methods: three Bayesian confidence intervals based on the Jeffreys’ (BAY-J), uniform (BAY-U), or normal-gamma-beta (BAY-NGB) prior and three highest posterior density intervals based on the Jeffreys’ (HPD-J), uniform (HPD-U), or normal-gamma-beta (HPD-NGB) prior.

Methods

Let Xi be a random variable following gamma (α, β) distribution with shape parameter α and scale parameter β. The probability density function can be derived as follows (1) fx;α,β=1Γaβαxα−1e−x/β;x>0,0;otherwise.

Suppose that the population of interest contains both zero and non-zero observations; the zero observations follow a binomial distribution while the non-zero observations follow a gamma distribution. The numbers of zero and non-zero observations are defined as n(0) and n(1) respectively, where n = n(0) + n(1). Let X = (X1, X2, …, Xn) be a random sample from a gamma distribution with excess zeros denoted as Δ(δ, α, β). The distribution function for the confidence interval can be derived as (2) Gxi;δ,α,β=δ;x=0,δ+1−δFx;α,β;x>0

where F(x; α, β) is the gamma cumulative distribution function.

The maximum likelihood estimator of δ is δ ^=n0/n. The population mean and variance of X are respectively given by (3) EX=1−δ⋅αβ

(4) VarX=τ=1−δ⋅αβ2+δ1−δ⋅αβ2.

The approches used to construct the confidence intervals are in the following subsections.

The FQ confidence interval

Krishnamoorthy, Mathew & Mukherjee (2008) suggested that a gamma distribution can be approximated by applying the cubic transformation of a Gaussian distribution. Let Y1, …, Yn be a sample from a gamma (α, β) distribution. When Xi=Yi13,i=1 , …, n then Xi are approximately normally distributed with mean µand variance σ2 respectively given by (5) μ=ba131−19aandσ2=b239a13

where shape parameter a and scale parameter b. The FQs for µand σ2 are, respectively, (6) Qμ=x ¯+Zn−1χn−12⋅snandQσ2=n−1s2χn−12

where x ¯ and s are the observed values of X ¯ and S, respectively; Z and χn−12 represent independent random variable of standard normal and chi-squared distribution, respectively; and n is the sample size. The FQs for the parameters of a gamma distribution can thus be derived as (7) Qa=191+0.5Qμ2Qσ2+1+0.5Qμ2Qσ22−112

(8) Qb=27Qa12Qσ232.

Krishnamoorthy & Wang (2016) proposed the FQs for the mean of gamma distribution as follows: (9) QM=Qμ2+Qμ22+Qσ23

where Qμ and Qσ2 are defined in Eq. (6). Li, Zhou & Tian (2013) proposed the FQ for δ as (10) Qδ∼12Betan1,n0+1+12Betan1+1,n0.

We can express the FQ for the variance as follows:

If V = ab2, then we can write Eq. (5) as (11) μ=V13b−131−b29Vandσ2=b439V13.

By solving the above equations for V, we obtain V=μ+μ2+4σ2/29−1/4σ2−1/44. Thus, the FQ for gamma variance can be obtained as (12) QV=Qμ+Qμ2+4Qσ229−1/4Qσ2−1/44

where Qμ and Qσ2 are defined in Eq. (6). Thus, the FQ for τ is in the form (13) Qτ=1−Qδ⋅QV+Qδ1−Qδ⋅QM2.

Therefore, the 100(1 − α)% confidence interval for τ is (14) CIFQ=Qτα/2,Qτ1−α/2

where Qτ(α/2) and Qτ(1 − α/2) are the (α/2)-th and (1 − α/2)-th percentiles of Qτ, respectively.

The confidence intervals for τ can be obtained by using Algorithm 1.

_______________________ Algorithm 1 FQ_____________________________________________________________________  1:  Generate x from a gamma distribution with excess zeros, compute ˉ x, and      s2 of the cube root transformed sample.   2:  Generate a standard normal variate Z and a chi-square variate χ2n−1.   3:  Generate Beta(n(1),n(0) + 1) and Beta(n(1) + 1,n(0).   4:  Compute Qμ, Qσ2  and Qδ from Eqs. (6) and (10).   5:  Compute the FQs for mean (QM) and variance (QV ) of gamma distribution      from Eqs. (9) and (12).   6:  Compute Qτ from Eq. (13).   7:  Repeat Steps 2–6 5,000 times and obtain an array of Qτ.   8:  Compute the 95% confidence intervals for τ from Eq. (14).   9:  Repeat Steps 1–8 10,000 times to compute the coverage probabilities (CPs)      and the average lengths (ALs).__________________________________________________________

The PB confidence interval

The log-likelihood function for the vector of shape α and scale β parameters in gamma distribution is given by Saulo et al. (2018). (15) Lα,β=nαlogβ−logΓα+α−1∑i=1n logXi−β∑i=1nXi.

Then, the maximum likelihood estimators (MLE) of α and β can be derived as (16) α ^=0.5logx ¯−logx¯

(17) β ^=α ^x ¯.

The PB for variance of gamma distribution with excess zeros can be written as (18) τ ^∗=1−δ ^∗⋅+δ ^∗1−δ ^∗⋅α ^∗β ^∗2.

The 100(1 − α)% confidence interval for τ is (19) CIPB=τ ^∗α/2,τ ^∗1−α/2.

The Bayesian confidence intervals

For this study, let Y1, …, Yn be a sample from a gamma (α, β) distribution, then for Xi=Yi13,i=1 , …, n then Xi are approximately normally distributed with mean µand variance σ2 (Krishnamoorthy, Mathew & Mukherjee, 2008). From the law of large numbers, we know that μ∼Nx ¯,σ2/n (Casella & Berger, 2001). Thus, the marginal posterior distribution of µis μ|σ2,x∼Nx ¯,σ2/n1

_______________________________________________________________________________________________________ Algorithm 2 PB_____________________________________________________________________  1:  Generate x from a gamma distribution with excess zeros, compute ˉ x, ^ δ , ^ α     and ^ β .   2:  Generate x∗ from x.   3:  Compute ˉ x∗, ^ δ∗, ^ α∗ and ^ β∗.   4:  Compute ^ τ∗ from Eq. (18).   5:  Repeat Steps 2–4 5,000 times and obtain an array of ^ τ∗.   6:  Compute the 95% confidence intervals for ^ τ∗ from Eq. (19).   7:  Repeat Steps 1–6 10,000 times to compute the CPs and ALs.__________________

HPD intervals are constructed from the posterior distribution based on the Bayesian approach. The HPD consists of the values of the parameter for which the posterior density is highest (Casella & Berger, 2001), while the HPD interval is the narrowest possible interval for the parameter of interest at probability 100(1 − α)% (Maneerat, Niwitpong & Niwitpong, 2020).

In this section, the Bayesian confidence interval is constructed upon the Jeffreys’ priors, uniform priors and normal-gamma-beta prior.

The BAY-J and HPD-J intervals

The Jeffreys’ prior for δ in a binomial distribution is pδ∝δ−121−δ12 (Bolstad & Curran, 2016). This leads to obtaining the marginal posterior distribution of δ as (20) δjef|x∼Betan0+12,n1+32.

Jeffreys’ prior for σ2 in a lognormal distribution is p(σ2) ∝ σ−2. Therefore, the marginal posterior distribution of σ2 becomes (21) σjef2|x∼IGn12,∑i=1nxi−μ22.

The marginal posterior distribution of µis (22) μjef|σ2,x∼Nx ¯,σjef2/n1.

We compute the mean and variance of gamma by using μjef|σ2, x and σjef2|x as follows: (23) MBAY−J=μjef2+μjef22+σjef23

(24) VBAY−J=μjef+μjef2+4σjef229−1/4σjef2−1/44.

So that (25) τ ^BAY−J=1−δjef⋅VBAY−J+δjef1−δjef⋅MBAY−J2.

The confidence interval and HPD interval of τ based on the Jeffreys’ prior are obtained as (26) CIBAY−J=τ ˆBAY−Jα/2,τ ˆBAY−J1−α/2.

The BAY-U and HPD-U intervals

The uniform prior for δ in a binomial distribution is p(δ) ∝ 1 (Bolstad & Curran, 2016). This leads to obtaining the marginal posterior distribution of δ as (27) δunif|x∼Betan0+1,n1+1.

The uniform prior for σ2 is σ2 ∝ 1 (Kalkur & Rao, 2017). Subsequently, the marginal posterior distribution of σ2 becomes (28) σunif2|x∼IGn1−22,∑i=1nxi−μ22.

The marginal posterior distribution of µas (29) μunif|σ2,x∼Nx ¯,σunif2/n1.

We compute the mean and variance of a gamma distribution using μunif|σ2, x and σunif2|x as follows: (30) MBAY−U=μunif2+μunif22+σunif23

(31) VBAY−U=μunif+μunif2+4σunif229−1/4σunif2−1/4.4

So that (32) τ ^BAY−U=1−δunif⋅VBAY−U+δunif1−δunif⋅MBAY−U2.

The confidence interval and HPD interval of τ based on the uniform prior are respectively obtained as (33) CIBAY−U=τ ˆBAY−Uα/2,τ ˆBAY−U1−α/2.

The BAY-NGB and HPD-NGB intervals

Maneerat & Niwitpong (2021) defined the normal-gamma-beta prior as (34) pτ∝λ−1δ1−δ−1/2

where λ = σ−2, (μ, λ) follows a normal-gamma distribution and δ follows a beta distribution (Maneerat & Niwitpong, 2021). Thus, the marginal posterior distributions of δ, σ2 and µrespectively become (35) δNGB|x∼Betan0+12,n1+12

(36) σNGB2|x∼IGn1−12,∑i=1n1xi−μ22

(37) μNGB|x∼t2n1−1x ¯,∑i=1nxi−x ¯2n1n1−1.

We compute the mean and variance of a gamma distribution by using μNGB|x and σNGB2|x as follows: (38) MBAY−NGB=μNGB2+μNGB22+σNGB23

(39) VBAY−NGB=μNGB+μNGB2+4σNGB229−1/4σNGB2−1/44.

So that (40) τ ^BAY−NGB=1−δNGB⋅VBAY−NGB+δNGB1−δNGB⋅MBAY−NGB2.

The confidence interval and HPD interval of τ based on the normal-gamma-beta prior are respectively obtained as (41) CIBAY−NGB=τ ˆBAY−NGBα/2,τ ˆBAY−NGB1−α/2.

_______________________________________________________________________________________________________ Algorithm 3 Bayesian interval___________________________________________________  1:  Generate x from a gamma distribution with excess zeros, compute ^ δ , ^ μ , and      ^ σ2.   2:  Generate δ|x from Eqs. (20), (27) and (35).   3:  Generate σ2|x from Eqs. (21), (28) and (36).   4:  Given σ2|x generate μ|σ2,x.   5:  Compute mean and variance of gamma distribution from Eqs.  (23), (24),      (30), (31), (38) and (39).   6:  Compute ^ τ from Eqs. (25), (32) and (40).   7:  Compute the 95% confidence intervals and HPD for ^ τ from Eqs. (26), (33)      and (41).   8:  Repeat Steps 1–7 10,000 times to compute the CPs and ALs.__________________

Simulation studies and Results

A Monte Carlo simulation study with 10,000 replications (M) and 5,000 repetitions (m) for FQ and PB, was conducted at a nominal confidence level of 0.95. We set sample size n as 30, 50, 100 or 200 and probability of zeros δ as 0.2, 0.5 or 0.8, for which we set shape parameter α as 7.00, 7.50 or 7.75; 2.00, 2.50 or 2.75; and 1.25, 1.50 or 1.75, respectively. We set rate parameter β as 1 for all cases. The performances of the confidence intervals were assessed by comparing their coverage probabilities (CPs) and average lengths (ALs); the best-performing confidence interval for a particular situation was identified as having a CP close or greater than 0.95 and the shortest AL. The confidence intervals for the variance of gamma distribution with excess zeros constructed using FQ, PB, BAY-J, HPD-J, BAY-U, HPD-U, BAY-NGB and HPD-NGB.

We report the coverage probabilities and the average lengths of nominal 95% two-sided confidence intervals for variance of gamma distribution with excess zeros in Table 1 and Figs. 1, 2 and 3.

Table 1 The coverage probabilities and (average lengths) of nominal 95% two-sided confidence intervals for variance of gamma distribution with excess zeros.

n	δ	α	Coverage probability (Average length)	
			PB	FQ	BAY-J	HPD-J	BAY-U	HPD-U	BAY-NGB	HPD-NGB	
30	0.2	7.00	0.9444	0.9686	0.9226	0.9184	0.9324	0.9444	0.9802	0.9771	
			(11.6924)	(12.6084)	(10.0751)	(9.8624)	(10.9472)	(10.6335)	(12.7642)	(12.4067)	
		7.50	0.9480	0.9728	0.9317	0.9293	0.9420	0.9522	0.9826	0.9789	
			(12.8866)	(13.6348)	(11.0851)	(10.8819)	(11.9674)	(11.6665)	(13.9564)	(13.6000)	
		7.75	0.9541	0.9731	0.9378	0.9334	0.9482	0.9569	0.9827	0.9807	
			(13.5974)	(14.3134)	(11.7150)	(11.5094)	(12.6155)	(12.3114)	(14.6896)	(14.3333)	
	0.5	2.00	0.8616	0.9521	0.8004	0.7817	0.8487	0.8557	0.9578	0.9391	
			(2.9978)	(4.1962)	(2.3918)	(2.0896)	(3.2420)	(2.7330)	(3.8034)	(3.3989)	
		2.50	0.8629	0.9500	0.7903	0.7788	0.8354	0.8502	0.9556	0.9440	
			(3.7780)	(5.3509)	(3.0529)	(2.7002)	(4.0796)	(3.4959)	(4.8638)	(4.4099)	
		2.75	0.8601	0.9467	0.7850	0.7767	0.8308	0.8433	0.9543	0.9454	
			(4.1300)	(5.8440)	(3.3308)	(2.9635)	(4.4162)	(3.8167)	(5.3407)	(4.8762)	
	0.8	1.25	0.7784	0.9564	0.8347	0.8479	0.8874	0.9569	0.9711	0.9616	
			(1.3763)	(12.5762)	(3.5742)	(2.0919)	(63.4518)	(15.9813)	(10.0543)	(4.5952)	
		1.50	0.7932	0.9615	0.8403	0.8577	0.8897	0.9603	0.9754	0.9671	
			(1.6638)	(13.2999)	(4.0138)	(2.4576)	(63.9289)	(6.6673)	(10.6502)	(5.1506)	
		1.75	0.8048	0.9621	0.8489	0.8647	0.8937	0.9637	0.9793	0.9725	
			(1.9395)	(12.9027)	(4.1595)	(2.7093)	(53.5498)	(15.2379)	(10.3024)	(5.4055)	
50	0.2	7.00	0.9621	0.9704	0.9275	0.9243	0.9411	0.9461	0.9814	0.9789	
			(9.2009)	(9.0634)	(7.5400)	(7.4561)	(7.8353)	(7.7315)	(9.4418)	(9.2934)	
		7.50	0.9625	0.9704	0.9338	0.9296	0.9447	0.9506	0.9807	0.9779	
			(10.1651)	(9.9058)	(8.3868)	(8.3065)	(8.6808)	(8.5812)	(10.4194)	(10.2715)	
		7.75	0.9655	0.9729	0.9374	0.9367	0.9463	0.9498	0.9844	0.9826	
			(10.6812)	(10.3530)	(8.8378)	(8.7599)	(9.1334)	(9.0356)	(10.9368)	(10.7863)	
	0.5	2.00	0.9054	0.9478	0.7868	0.7573	0.8238	0.8155	0.9505	0.9285	
			(2.4797)	(2.6883)	(1.6201)	(1.4938)	(1.8801)	(1.7160)	(2.5473)	(2.3981)	
		2.50	0.9010	0.9475	0.7890	0.7693	0.8228	0.8202	0.9514	0.9341	
			(3.0615)	(3.4346)	(2.0567)	(1.9090)	(2.3755)	(2.1861)	(3.2687)	(3.1047)	
		2.75	0.9039	0.9515	0.7892	0.7674	0.8223	0.8193	0.9538	0.9417	
			(3.3850)	(3.8265)	(2.2825)	(2.1243)	(2.6329)	(2.4295)	(3.6435)	(3.4714)	
	0.8	1.25	0.8435	0.9559	0.8337	0.8262	0.8882	0.9116	0.9688	0.9476	
			(1.1826)	(2.4727)	(1.2830)	(1.0168)	(2.5640)	(1.7317)	(2.1219)	(1.6296)	
		1.50	0.8550	0.9569	0.8384	0.8402	0.8860	0.9161	0.9699	0.9538	
			(1.4185)	(2.8663)	(1.5275)	(1.2448)	(2.8649)	(2.0242)	(2.4800)	(1.9666)	
		1.75	0.8675	0.9602	0.8515	0.8537	0.8930	0.9239	0.9736	0.9622	
			(1.6807)	(3.2832)	(1.7911)	(1.4943)	(3.1990)	(2.3387)	(2.8757)	(2.3349)	
100	0.2	7.00	0.9685	0.9652	0.9270	0.9238	0.9372	0.9366	0.9758	0.9729	
			(6.6077)	(6.1266)	(5.2494)	(5.2171)	(5.3267)	(5.2916)	(6.5195)	(6.4617)	
		7.50	0.9732	0.9682	0.9321	0.9311	0.9394	0.9407	0.9801	0.9785	
			(7.3264)	(6.7357)	(5.8730)	(5.8403)	(5.9473)	(5.9122)	(7.2284)	(7.1679)	
		7.75	0.9760	0.9702	0.9437	0.9426	0.9501	0.9512	0.9841	0.9817	
			(7.6438)	(7.0104)	(6.1733)	(6.1407)	(6.2465)	(6.2117)	(7.5695)	(7.5095)	
	0.5	2.00	0.9332	0.9292	0.7597	0.7285	0.7931	0.7636	0.9316	0.9120	
			(1.8169)	(1.6738)	(1.0400)	(0.9967)	(1.1103)	(1.0616)	(1.6412)	(1.5930)	
		2.50	0.9360	0.9420	0.7703	0.7434	0.7995	0.7783	0.9436	0.9306	
			(2.2541)	(2.1692)	(1.3337)	(1.2817)	(1.4222)	(1.3641)	(2.1292)	(2.0751)	
		2.75	0.9301	0.9392	0.7763	0.7528	0.7995	0.7875	0.9425	0.9295	
			(2.4789)	(2.4163)	(1.4761)	(1.4200)	(1.5739)	(1.5114)	(2.3735)	(2.3171)	
	0.8	1.25	0.9076	0.9439	0.8161	0.7969	0.8573	0.8475	0.9550	0.9302	
			(0.9191)	(1.0226)	(0.6335)	(0.5746)	(0.7685)	(0.6809)	(0.9717)	(0.8766)	
		1.50	0.9159	0.9526	0.8333	0.8141	0.8667	0.8630	0.9624	0.9427	
			(1.0920)	(1.2439)	(0.7821)	(0.7184)	(0.9349)	(0.8409)	(1.1916)	(1.0887)	
		1.75	0.9123	0.9544	0.8394	0.8267	0.8696	0.8697	0.9678	0.9482	
			(1.2881)	(1.4819)	(0.9445)	(0.8765)	(1.1158)	(1.0159)	(1.4312)	(1.3194)	
200	0.2	7.00	0.9761	0.9634	0.9225	0.9199	0.9339	0.9330	0.9751	0.9715	
			(4.7169)	(4.2392)	(3.6845)	(3.6666)	(3.7070)	(3.6888)	(4.5589)	(4.5303)	
		7.50	0.9785	0.9665	0.9317	0.9304	0.9442	0.9428	0.9775	0.9755	
			(5.1932)	(4.6428)	(4.1173)	(4.0987)	(4.1390)	(4.1201)	(5.0479)	(5.0179)	
		7.75	0.9822	0.9692	0.9403	0.9384	0.9483	0.9469	0.9817	0.9799	
			(5.4485)	(4.8598)	(4.3497)	(4.3307)	(4.3699)	(4.3503)	(5.3069)	(5.2765)	
	0.5	2.00	0.9477	0.8978	0.6997	0.6659	0.7285	0.6938	0.9000	0.8774	
			(1.3034)	(1.1146)	(0.7016)	(0.6854)	(0.7237)	(0.7066)	(1.1126)	(1.0944)	
		2.50	0.9463	0.9201	0.7363	0.7060	0.7590	0.7326	0.9209	0.9059	
			(1.6261)	(1.4556)	(0.9051)	(0.8852)	(0.9330)	(0.9121)	(1.4510)	(1.4304)	
		2.75	0.9470	0.9297	0.7443	0.7145	0.7667	0.7419	0.9302	0.9162	
			(1.7859)	(1.6291)	(1.0051)	(0.9835)	(1.0358)	(1.0131)	(1.6250)	(1.6031)	
	0.8	1.25	0.9426	0.9268	0.7829	0.7506	0.8168	0.7872	0.9383	0.9111	
			(0.6664)	(0.5858)	(0.3827)	(0.3651)	(0.4131)	(0.3922)	(0.5860)	(0.5575)	
		1.50	0.9476	0.9450	0.8173	0.7885	0.8451	0.8224	0.9553	0.9334	
			(0.8008)	(0.7339)	(0.4859)	(0.4665)	(0.5214)	(0.4983)	(0.7377)	(0.7060)	
		1.75	0.9462	0.9480	0.8317	0.8124	0.8542	0.8393	0.9594	0.9419	
			(0.9469)	(0.8890)	(0.5953)	(0.5742)	(0.6363)	(0.6114)	(0.8986)	(0.8637)	
Notes.

a The coverage probabilities greater than the nominal confidence level of 0.95 are in bold and the shortest average lengths are in italics.

The CPs of the PB, FQ, HPD-U, BAY-NGB, and HPD-NGB confidence intervals were greater than or close to the nominal confidence level of 0.95 in all situations studied. For a small-to-moderate sample size, FQ and the HPD-U performed well for small δ whereas BAY-NGB and HPD-NGB performed well for large δ. For a large sample size, FQ performed well for small δ whereas BAY-NGB performed well for large δ. Although the expected lengths of the HPD-J were shorter than the other methods, the CPs of BAY-J and HPD-J were lower than the nominal confidence level in all cases.

The findings show that although FQ, HPD-U, BAY-NGB, and HPD-NGB attained acceptable CPs, the ALs of BAY-NGB and the HPD-NGB were shorter than the other methods, and so they can be recommended for constructing the confidence interval for the variance of a gamma distribution with excess zeros. It can be seen that for HPD-NGB developed from the study of Maneerat & Niwitpong (2021), the simulation results are similar to these studies. For small-to-large sample size, HPD-NGB performed well. BAY-NGB and HPD-NGB are the best because BAY-NGB and HPD-NGB attained stable CPs and ALs were shorter than the other methods for all sample sizes. A referee suggested to check the validity and robustness of the model for smaller sample sizes with moderate number of zeros. We, therefore, simulated a study with 10,000 replications (M) and 5,000 repetitions (m) for FQ and PB, was conducted at a nominal confidence level of 0.95. We set sample size n as 10 or 20 and probability of zeros δ as 0.2, or 0.5, for which we set shape parameter α as 7.00, 7.50 or 7.75; and 2.00, 2.50 or 2.75, respectively. We set rate parameter β as one for all cases. The results (not shown here) show that the CPs of the FQ, HPD-U, BAY-NGB, and HPD-NGB confidence intervals were greater than or close to the nominal confidence level of 0.95 in all situations studied. The findings show that although FQ, HPD-U, BAY-NGB, and HPD-NGB attained acceptable CPs, the ALs of HPD-NGB were shorter than the other methods. Although the sample sizes are small (n = 10, n = 20), our findings show that BAY-NGB and HPD-NGB can be recommended for constructing the confidence interval for the variance of a gamma distribution with excess zeros.

Figure 1 Line graphs of (A) coverage probabilities and (B) average lengths of all methods in the case of the different sample sizes.

Figure 2 Line graphs of (A) coverage probabilities and (B) average lengths of all methods in the case of the different probabilities of zero values.

Figure 3 Line graphs of (A) coverage probabilities and (B) average lengths of all methods in the case of the different shape parameters.

Empirical application of the proposed confidence intervals

The confidence interval performances were compared by using real-world datasets comprising monthly rainfall data reported by the Upper Northern Region Irrigation Hydrology for January and February 1993 to 2021 at the Kiew Lom Dam, Lampang province, Thailand.

First, the best fit for the positive rainfall data among normal, lognormal, Cauchy, and gamma models was examined by calculating their Akaike information criterion (AIC) and Bayesian information criterion (BIC) values (Table 2). The results show that the lowest AIC and BIC values (207.7139 and 210.2301, respectively) were for the gamma distribution, indicating that it was the best fit for the data.

The summary statistics for the rainfall data in Kiew Lom Dam Lampang province are x ¯=18.6461, n = 58 , n(1) = 26, n(0) = 32, while the maximum likelihood estimators for δ, α, β and τ are δ ˆ=0.5517,α ˆ=0.7297,β ˆ=0.0391 and τ ˆ=299.5542, respectively. The calculated two-sided confidence intervals for τ are reported in Table 3.

Table 2 AIC and BIC results of positive rainfall data.

Models	Normal	Lognormal	Cauchy	Gamma	
AIC	224.9317	216.186	230.4221	207.7139	
BIC	227.4479	218.7022	232.9383	210.2301	

Table 3 The 95% two-sided confidence intervals for variance of rainfall data in Kiew Lom Dam in Lampang province.

Methods	Confidence intervals for θ	Length of intervals	
	Lower	Upper		
PB	115.6468	543.4372	427.7903	
FQ	115.3533	974.3039	858.9506	
BAY-J	138.7433	764.2119	625.4687	
HPD-J	107.4391	613.0399	505.6008	
BAY-U	146.4527	1078.71	932.2578	
HPD-U	111.2196	809.1257	697.9061	
BAY-NGB	135.4990	885.4536	749.9546	
HPD-NGB	102.1386	685.1513	583.0128	

For n = 50 and δ = 0.5, FQ and BAY-NGB obtained CPs close to the nominal confidence level of 0.95, but BAY-NGB bay obtained the shortest length method. Thus, the BAY-NGB method is recommended for constructing the confidence interval for the variance in rainfall data in January and February at the Kiew Lom Dam in Lampang province.

Conclusions

We constructed confidence intervals for the variance of a gamma distribution with excess zeros by using the PB, FQ, BAY-J, HPD-J, BAY-U, HPD-U, BAY-NGB, and HPD-NGB approaches. The CPs and ALs of the methods were assessed by Monte Carlo simulation for various situations and by using real precipitation data following a gamma distribution with excess zeros. Our findings show that BAY-NGB and HPD-NGB can be recommended for constructing the confidence interval for the variance of a gamma distribution with excess zeros. In future research, we will investigate constructing confidence intervals for the difference between the variances of gamma distributions with excess zeros.

Supplemental Information

Supplemental Information 1 R code

This code computed coverrage probabilities and average lengths for all confidence intervals.

Click here for additional data file.

Supplemental Information 2 R code to compute the data set

This R code is computed all confidence intervals

Click here for additional data file.

Supplemental Information 3 The monthly rainfall data (mm) from Kiew Lom Dam, Lampang province, Thailand in January and February, comprising 58 observations from 1993–2021

Click here for additional data file.

Additional Information and Declarations

Competing Interests

Author Contributions

Data Availability

The authors declare there are no competing interests.

Wansiri Khooriphan conceived and designed the experiments, performed the experiments, analyzed the data, prepared figures and/or tables, authored or reviewed drafts of the article, and approved the final draft.

Sa-Aat Niwitpong conceived and designed the experiments, analyzed the data, authored or reviewed drafts of the article, and approved the final draft.

Suparat Niwitpong conceived and designed the experiments, performed the experiments, analyzed the data, authored or reviewed drafts of the article, and approved the final draft.

The following information was supplied regarding data availability:

The raw data are available as Supplemental File.

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
