# Peer review of "Bayesian estimation of rainfall dispersion in Thailand using gamma distribution with excess zeros"

_PeerJ, doi:10.7717/peerj.14023_

## Round 0.1 · original submission · Minor Revisions

Overall, the discussion part is weak. The Discussion should summarize the manuscript's main finding(s) in the context of the broader scientific literature and address any limitations of the study or results that conflict with other published work.

Reviewer 1 ·

Basic reporting

The text is written clearly. The authors say what they want to do.

Experimental design

The data are coming from routinely collected data and this should be fine.

Validity of the findings

I think the findings are plausible and make sense

Additional comments

Comments for the author:
I think the paper is fine. I would like to ask the authors a few questions if I may.
1. In abstract, the sentence “Rainfall dispersion is important and interesting, it can be predicted using the confidence intervals for the variance of a gamma distribution with excess zeros.”. In Statistical inference, a CI for a fixed parameter represents a plausible range of value for the parameter that is consistent with the observed data. The authors need to explain why the CIs for the variance of a gamma model with excess zeros can be used to predict. Please clarify.

2. In Bayesian credible and highest posterior density intervals, there are many prior distributions, i.e., Jeffrey’s, uniform, and normal-gamma-beta priors. How do the authors select the prior in real situations? The author should mention in this matter.

3. It should be pointed out what is your motivation to propose the same priors as the research article “Bayesian approach to construct confidence intervals for comparing the rainfall dispersion in Thailand” published in PeerJ. Please explain.

4. In the presentation of simulation results: it is not clear from tables or graphs which the best-performing method is. I spend more time illuminating this.

5. In empirical application, the authors do this for a very select dataset. If this is done routinely per day or per month, which method should be used taking into account that it must be practicable?

6. The author should discuss the simulation results and why the BAY-NGB and HPD-NGB are the best (lines 148-149). This could give the readers a clearer idea.

Minors:
1. In Table 1, the boldface should be stated what does it mean?
2. The authors should use capitalization at the beginning of a word, i.e., Table 3. Please carefully check throughout the manuscript.

Reviewer 2 ·

Basic reporting

No Comment.

Experimental design

No comment.

Validity of the findings

Validity of the model could have been checked for smaller sample sizes with moderate number of zeros. All the sample sizes are simulated was 30 or more. The robustness of the model could have been checked using small sample sizes such as 10, 20 etc with possible proportion of zeros, delta = .3, .5 etc. It would be interesting to see how these methods perform under those conditions.

Additional comments

The article is well written, clear and easy to understand. The Rcodes are easy to be checked and verified. It would be interesting to see how your methods are performing when the sample sizes are smaller.

Reviewer 3 ·

Basic reporting

The paper deals with an important problem -- providing confidence intervals for gamma distribution with zeros -- which is the best description of rainfall in Thailand. The paper compares different methods and concludes which method to use when. It is well written, with adequate references.

Experimental design

The authors apply several known methods to observations data.

Validity of the findings

Conclusions are well-justified and useful

Additional comments

Minor editing is needed:

abstract: intervalsfor -> intervals for

formula (5): use \left( and \right) in LaTeX, to make sure that the parentheses cover everything in between; same in (11), (20), (21), and (35)

formulas (15) and (16): log in LaTeX should be \log

---

## Round 0.2 · accepted · Accept

I congratulate the authors for the effort put into this paper! The manuscript is significantly improved; therefore, I recommend accepting it in its current form!

Reviewer 1 ·

Basic reporting

The text is well and clear in the revision

Experimental design

No comment

Validity of the findings

No comment

Additional comments

This revision is clearer than the submitted manuscript. So, I have recommended this revision published in PeerJ.